# Learning Large Neighborhood Search Policy for Integer Programming

**Yaoxin Wu**
SCALE@NTU Corp Lab
Nanyang Technological University, Singapore
`wuyaoxin@ntu.edu.sg`

**Wen Song**[*]
Shandong University
Qingdao, China
`wensong@email.sdu.edu.cn`

**Zhiguang Cao**
Singapore Institute of Manufacturing Technology
A*STAR, Singapore
`zhiguangcao@outlook.com`

**Jie Zhang**
Nanyang Technological University
Singapore
`zhangj@ntu.edu.sg`

## Abstract

We propose a deep reinforcement learning (RL) method to learn large neighborhood search (LNS) policy for integer programming (IP). The RL policy is trained as the destroy operator to select a subset of variables at each step, which is reoptimized by an IP solver as the repair operator. However, the combinatorial number of variable subsets prevents direct application of typical RL algorithms. To tackle this challenge, we represent all subsets by factorizing them into binary decisions on each variable. We then design a neural network to learn policies for each variable in parallel, trained by a customized actor-critic algorithm. We evaluate the proposed method on four representative IP problems. Results show that it can find better solutions than SCIP in much less time, and significantly outperform other LNS baselines with the same runtime. Moreover, these advantages notably persist when the policies generalize to larger problems. Further experiments with Gurobi also reveal that our method can outperform this state-of-the-art commercial solver within the same time limit.

## 1 Introduction

Combinatorial optimization problems (COPs) have been widely studied in computer science and operations research, which cover numerous real-world tasks in many fields such as communication, transportation and manufacturing [1]. Most COPs are very difficult to solve efficiently due to their NP-hardness. The performance of classic methods, including exact and heuristic algorithms [2], is generally limited by hand-crafted policies that are costly to design, since considerable trial-and-error and domain knowledge are needed. On the other hand, it is common in practice that similar instances with shared structure are frequently solved, and differ only in data that normally follows a distribution [3]. This provides a chance for machine learning to *automatically* generate heuristics or policies. In doing so, the learned alternatives are expected to save massive manual work in algorithm design, and raise the performance of the algorithm on a class of problems.

Recently, a number of works apply deep (reinforcement) learning to automatically design heuristic algorithms, either in constructive or improving fashion. Different from construction heuristics that sequentially extend partial solutions to complete ones [4, 5, 6, 7, 8, 9, 10, 11], learning *improvement heuristics* can often deliver high solution quality by iteratively reoptimizing an initial solution using

---

[*]Wen Song is the corresponding author.

35th Conference on Neural Information Processing Systems (NeurIPS 2021).

local operations [12, 13, 14]. In this line, some methods are developed under the Large Neighborhood Search (LNS) framework [15, 16], which is a powerful improving paradigm to find near-optimal solutions for COPs.

However, the above methods are restricted to specific problem types, and cannot generalize to those from different domains. This motivates the studies of learning to directly solve Integer Programs (IPs), which is very powerful and flexible in modelling a wide range of COPs. The standard approach to solve IPs is branch-and-bound (B&B) [17], which lies at the core of common solvers such as SCIP, Gurobi, and CPLEX. Thus, most of existing methods improve the performance of a solver on a distribution of instances, by training models for critical search decisions in B&B such as variable and node selection [18, 19, 20]. Nevertheless, these methods are generally limited to small instances and require sufficient interface access to the internal solving process of the solvers.

This paper mainly tackle the issue that *how to improve a solver from externals such that it can find high-quality solutions more quickly?* In specific, we propose a high-level, learning based LNS method to solve general IP problems. Based on deep reinforcement learning (RL), we train a policy network as the destroy operator in LNS, which decides a subset of variables in the current solution for reoptimization. Then we use a solver as the repair operator, which solves sub-IPs to reoptimize the destroyed variables. Despite being heuristic, our method can effectively handle the large-scale IP by solving a series of smaller sub-IPs. Moreover, complex interface to the solver's internal logic is not required. However, the above RL task is challenging, mainly because the action space, i.e., number of variable subsets at each LNS step, is exponentially large. To resolve this issue, we represent all the subsets by factorizing them into binary decisions on each variable, i.e., whether a variable should be destroyed. In doing so, we make it possible to learn a policy to select *any subset* from large discrete action spaces (at least $2^{1000}$ candidates in our experiments). To this end, we design a Graph Neural Network (GNN) based policy network that enables learning policies for each variable in parallel, and train it by a customized actor-critic algorithm.

A recent work [21] also attempts to learn LNS policy to solve IP problems, and we generalize this framework to enable learning more flexible and powerful LNS algorithms. One limitation of [21] is that it hypothesizes a constant cardinality of the destroyed variable subset at each LNS step, which is a predefined hyperparameter. However, the number and choice of optimized variables at each step should be adaptive according to instance information and solving status, which is achieved in our method. In doing so, the LNS policies trained by our method significantly outperform those trained by the method in [21].

We evaluate our method on four NP-hard benchmark problems with SCIP as the repair solver. Extensive results show that our method generally delivers better solutions than SCIP with mere 1/2 or 1/5 of runtime, and significantly outperforms LNS baselines with the same runtime. These advantages notably persist when the trained policies are directly applied to much larger problems. We also apply our LNS framework to Gurobi, which shows superiority over the solver itself and other baselines.

## 2  Related work

In this section, we briefly review existing works related to ours. We first describe two main streams of learning based methods to solve COPs, and then introduce the literature that study RL with large discrete action space, which is also an essential issue we confront in this paper.

**Learning to solve specific COPs.**  Quite a few works attempt to learn heuristics to solve certain types of COPs. Compared to construction ones, methods that learn improvement heuristics can often deliver smaller optimality gap, by training policies to iteratively improve the solution. Chen and Tian [12] propose to learn how to locally rewrite a solution; Wu et al. [13] train policies to pick the next solution in local moves; Lu et al. [14] learn to select local operators to reform a solution. These methods are generally limited by simple local operations. A few methods learn more powerful operators under the LNS framework. Hottung and Tierney [15] train an attention model to repair the solution every time it is broken by a predefined destroy operator. Similarly, Gao et al. [16] combine GNN and Recurrent Neural Network to learn a reinsertion operator to repair sequence-based solution. However, all the above methods are limited to specific problem types, e.g., the LNS methods in [15, 16] are designed only for routing problems. In contrast, this paper aims to solve general IP problems with a high-level LNS framework and raise its performance by learning better policies.

**Learning to solve IP problems.** Most of learning based methods for IPs aim to improve inner policies of B&B algorithm. For example, He et al. [18] learn to explore nodes in B&B tree by imitating an oracle. Gasse et al. [20] train a GNN model to predict the strong branching rule by imitation learning. Khalil et al. [19] predict the use of primal heuristics by logistic regression. Other components of B&B are also considered to improve its practical performance, such as learning to select cutting planes by Tang et al. [22], predicting local branching at root node by Ding et al. [23] or refining the primal heuristic and branching rule concurrently by Nair et al. [24]. Different from these works, we employ RL to improve practical performance of IP solvers especially for large-scale problems, without much engineering effort on interfacing with inner process of solvers.[2] This is also noted in [21], which proposes to combine learning and LNS to solve IPs. However, a major drawback of this method is that the subsets of destroyed variables are assumed to be fixed-sized, which limits its performance. In contrast, our LNS framework allows picking variable subsets in a more adaptive way, and thus empowers learning broader classes of policies.

**RL with large action spaces.** Learning with large-sized, high-dimensional discrete action spaces is still intricate in current RL research. In this direction, Pazis and Parr [25] use binary format to encode all actions, and learn value functions for each bit. Tavakoli et al. [26] design neural networks with branches to decide on each component of the action. Besides, Dulac-Arnold et al. [27] and Chandak et al. [28] update the action representation by solving continuous control problems so as to alleviate learning complexity via generalization. Other similar works can be found in [29, 30, 31]. However, we note that all these methods are generally designed for tasks with (discretized) continuous action spaces, which are relatively low-dimensional and small-sized (at most $10^{25}$ as in [26]). In contrast, action spaces in our scenario are much larger (at least $2^{1000}$). In this paper, we propose to factorize the action space into binary actions on each dimension, and train the individual policies in parallel through parameter sharing.

## 3 Preliminaries

**Integer Program (IP)** is typically defined as $\arg\min_x\{\mu^\top x | Ax \le b; x \ge 0; x \in \mathbb{Z}^n\}$, where $x$ is a vector of $n$ decision variables; $\mu \in \mathbb{R}^n$ denotes the vector of objective coefficients; the incidence matrix $A \in \mathbb{R}^{m \times n}$ and right-hand-side (RHS) vector $b \in \mathbb{R}^m$ together define $m$ linear constraints. With the above formulation, the size of an IP problem is generally reflected by the number of variables ($n$) and constraints ($m$).

**Large Neighborhood Search (LNS)** is a type of improvement heuristics, which iteratively reoptimizes a solution by the *destroy* and *repair* operator until certain termination condition is reached [32]. Specifically, the former one breaks part of the solution $x_t$ at step $t$, then the latter one fixes the broken solution to derive the next solution $x_{t+1}$. The destroy and repair operator together define the solution neighborhood $\mathcal{N}(x_t)$, i.e., solution candidates that can be accessed at time step $t$. Compared to typical local search heuristics, LNS is more effective in exploring the solution space, since a larger neighborhood is considered at each step [32]. Most of existing LNS methods rely on problem specific operators, e.g., removal and reinsertion for solutions with sequential structure [33, 34]. Instead, we propose a RL based LNS for general IP problems, with the learned destroy operator to select variables for reoptimization at each step.

## 4 Methodology

In this section, we first formulate our LNS framework as a Markov Decision Process (MDP). Then, we present the factorized representation of the large-scale action space, and parametrize the policy by a specialized GNN. Finally, we introduce a customized actor-critic algorithm to train our policy network for deciding the variable subset.

### 4.1 MDP formulation

Most of existing works learn LNS for specific problems [15, 16], which rely on extra domain knowledge and hinder their applicability to other COPs. In this paper, we apply LNS to general IP

---

[2]Nevertheless, our method can also work with solvers enhanced by the above methods as repair operators.

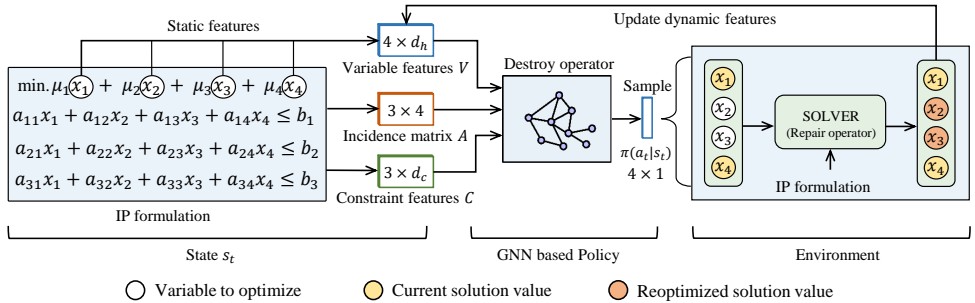

Figure 1: **An example of LNS solving an instance with 4 variables and 3 constraints.** Given the current state characterized by static and dynamic features, the RL agent (destroy operator) selects a subset of variables from the current solution to be reoptimized. Then this action influences the environment, through which a sub-IP is formulated and solved by the repair operator (i.e., solver). As feedbacks, the new solution updates the state and a reward is provided to the agent. The above process is repeated until reaching the time limit.

problems, in which RL is employed to learn a policy that at each step, selects a variable subset from the solution to be reoptimized. We formulate this sequential decision problem as a discrete-time MDP. Specifically, we regard the destroy operator as the agent and the remainder in LNS as the environment. In one episode of solving an IP instance, the MDP can be described as follows:

**States.** A state $s_t \in \mathcal{S}$ at each LNS step needs to not only reflect the instance information but also the dynamic solving status. For the former, we represent it by static features of variables and constraints, along with the incidence matrix in an instance. The latter is represented by dynamic features, including the current solution $x_t$ and dynamic statistics of incumbent solutions up to step $t$. Specifically, both the current and incumbent solution at $t = 0$ are defined as the initial solution $x_0$.

**Actions.** At each state $s_t$, an action of the agent is to select a variable subset $a_t$ from all candidate subsets $\mathcal{A}$ for reoptimization.

**Transition.** The repair operator (i.e., an IP solver) solves the sub-IP, where only the variables in action $a_t$ are optimized with the others equaling to their current values in $x_t$. Accordingly, the next state $s_{t+1}$ is deterministically attained by updating $s_t$ with the new solution:

$$x_{t+1} = \arg\min_{x}\{\mu^\top x | Ax \le b; x \ge 0; x \in \mathbb{Z}^n; x^i = x_t^i, \forall x^i \notin a^t\}. \tag{1}$$

**Rewards.** The reward function is defined as $r_t = r(s_t, a_t) = \mu^\top(x_t - x_{t+1})$, which is the change of the objective value. Given the step limit $T$ of the interaction between the agent and environment, the return (i.e., cumulative rewards) from step $t$ of the episode is $R_t = \sum_{k=t}^{T} \gamma^{k-t} r_k$, with the discount factor $\gamma \in [0, 1]$. The goal of RL is to maximize the expected return $\mathbb{E}[R_1]$ over all episodes, i.e., the expected improvement over initial solutions, by learning a policy.

**Policy.** The stochastic policy $\pi$ represents a conditional probability distribution over all possible variable subsets given a state. Starting from $s_0$, it iteratively picks an action $a_t$ based on the sate $s_t$ at each step $t$, until reaching the step limit $T$.

An example of the proposed LNS framework solving an IP instance is illustrated in Figure 1, in which the GNN based policy network has the potential to process instances of any size via sharing parameters across all variables.

## 4.2 Action factorization

Given the vector of decision variables $x$ in an IP instance, we gather all its elements in a set $X = \{x^1, x^2, \dots, x^n\}$. Accordingly, we define the action space $\mathcal{A} = \{a | a \subseteq X\}$, which contains all possible variable subsets. Thus our RL task is learning a policy to select $a_t \in \mathcal{A}$ for reoptimization at each step $t$ of LNS, and the cardinality of $a_t$ (i.e., $|a_t|$) reflects the destroy degree on the solution. Apparently, the size of the combinatorial space $\mathcal{A}$ is $2^n$, which grows exponentially with the number

of variables $n$. This prevents the application of general RL algorithms on large-scale problems, since they require an explicit representation of all actions and exploration over such huge action space.

As a special case under our LNS framework, Song et al. [21] assumes the action space as the subspace of $\mathcal{A}$ that merely contains equal-sized variable subsets, that is, $\mathcal{A}_z = \{a_z | a_z \in \mathcal{A}, |a_z| = z\}$. In doing so, they instead learn classifying variables into groups of equal size, to optimize each iteratively. Despite the good performance on some COPs, such action representation with the fixed destroy degree makes the LNS search inflexible and thus limits the class of policies that can be learned. Also, the issue of large action space still exists, since the patterns to group variables could be combinatorial.

To involve larger action space and in the meanwhile keep the learning tractable, we factorize the combinatorial action space $\mathcal{A}$ into elementary actions on each dimension (i.e., variable). Specifically, we denote $a_t^i \in \{x^i, \emptyset\}$ as the elementary action for variable $x^i$ at step $t$. It means that $x^i$ is either selected for reoptimization, or not selected and thus fixed as its value in the current solution. With such representation, any variable subset can also be expressed as $a_t = \bigcup_{i=1}^n a_t^i$. Therefore, our task can be converted into learning policies for binary decisions on each variable. This enables exploration of large action space by traversing binary action spaces, the number of which grows linearly with the number of variables. To this end, we factorize the original policy as below:

$$\pi(a_t|s_t) = \prod_{i=1}^n \pi^i(a_t^i|s_t), \tag{2}$$

which expresses $\pi(a_t|s_t)$, the probability of selecting an action, as the product of probabilities of selecting its elements. Since $\sum_{a_t^i \in \{x^i, \emptyset\}} \pi^i(a_t^i|s_t) = 1$, $\forall i \in \{1, \ldots, n\}$, it is easy to verify that the sum of probabilities of all actions in $\mathcal{A}$ still equals to 1, i.e., $\sum_{a_t \in \mathcal{A}} \pi(a_t|s_t) = 1$. Based on this factorization, we can apply RL algorithms to train policies $\pi^i$ for each variable $x^i$. In the following subsections, we first parametrize the policy $\pi$ by a GNN $\pi_\theta$, and then train it by a customized actor-critic algorithm.

## 4.3  Policy parametrization

Policy networks in deep RL algorithms are generally designed to map states to probabilities of selecting each action. In our case, the complexity of directly training such policy networks could exponentially increase with the growing number of variables. Based on Equation (2), an ad hoc way to reduce the complexity is training individual policies for each dimension. However, the IP problems in our study comprise a high volume of variables, and it is unmanageable to train and store this many networks. Learning disjoint policies without coordination could also suffer from convergence problem as shown in [35]. Another possible paradigm could be constructing a semi-shared network, in which all dimensions share a base network and derive outputs from separate sub-networks. Following a similar idea, Tavakoli et al. [26] design a network with branches for deep Q-learning algorithms. However, it is not suitable for IP problems, since the number of sub-networks is only predefined for single problem size, without the generalization ability to different-sized instances.

To circumvent the above issues, we design a GNN based policy network to share parameters across all dimensions, and also enable generalization to any problem size [36, 37]. To this end, we first describe the state $s_t$ by a bipartite graph $\mathcal{G} = (\mathcal{V}, \mathcal{C}, \mathbf{A})$, where $\mathcal{V} = \{v_1, \cdots, v_n\}$ denotes variable nodes with features $\mathbf{V} \in \mathbb{R}^{n \times d_v}$; $\mathcal{C} = \{c_1, \cdots, c_m\}$ denotes constraint nodes with features $\mathbf{C} \in \mathbb{R}^{m \times d_c}$; $\mathbf{A} \in \mathbb{R}^{m \times n}$ denotes the adjacency matrix with $a_{ji}$ being the weight (or feature) of the edge between nodes $c_j$ and $v_i$, which is practically the incidence matrix $A$ in an IP instance. This state representation is similar to the one in [20], which depicts the status in B&B tree search to learn branching policy. We extend its usage outside the solver to reflect the solving process in our LNS framework.

Then, we parametrize the policy as $\pi_\theta(a_t|s_t)$ by a graph convolutional network (GCN), a GNN variant broadly used in various tasks [38, 39, 40]. The architecture of our design is illustrated in Appendix A.1, with the graph convolution layer expressed as below:

$$\begin{aligned} \mathbf{C}^{(k+1)} &= \mathbf{C}^{(k)} + \sigma \left( \text{LN} \left( \mathbf{A} \mathbf{V}^{(k)} W_v^{(k)} \right) \right), \\ \mathbf{V}^{(k+1)} &= \mathbf{V}^{(k)} + \sigma \left( \text{LN} \left( \mathbf{A}^\top \mathbf{C}^{(k+1)} W_c^{(k)} \right) \right), k = 0, \ldots, K \end{aligned} \tag{3}$$

where $W_v^{(k)}, W_c^{(k)} \in \mathbb{R}^{d_h \times d_h}$ are trainable weight matrices in the $k$-th layer; $\mathbf{V}^{(k)} = [\mathbf{v}_1^{(k)} \cdots \mathbf{v}_n^{(k)}]^\top$ and $\mathbf{C}^{(k)} = [\mathbf{c}_1^{(k)} \cdots \mathbf{c}_m^{(k)}]^\top$ are node embeddings for variables and constraints respectively in $k$-th

layer; LN and $\sigma(\cdot)$ denote layer normalization and Tanh activation function respectively. In particular, we linearly project raw features of variables and constraints into the initial node embeddings $\mathbf{V}^{(0)}$ and $\mathbf{C}^{(0)}$ with $d_h$ dimensions ($d_h = 128$), and keep this dimension through all layers. After $K$ iterations of convolution ($K = 2$), the embeddings for the two clusters of heterogeneous nodes are advanced as $\{\mathbf{v}_i^K\}_{i=1}^n$ and $\{\mathbf{c}_j^K\}_{j=1}^m$. We finally process the former by a multi-layer perceptron (MLP) with a single-value output activated by Sigmoid. In this way, the output value can represent the probability of a variable being selected, i.e., $\pi^i(a_t^i|s_t) = \pi_\theta(a_t^i|s_t) = \text{MLP}(\mathbf{v}_i^K), \forall i \in \{1, \ldots, n\}$, such that we can conduct Bernoulli sampling accordingly on each variable and attain the subset. In this paper, we structure the MLP by two hidden layers, which have 256 and 128 dimensions respectively and are activated by Tanh.

## 4.4 Training algorithm

Actor-critic is one of the policy gradient methods developed from REINFORCE [41]. It parametrizes state-value or action-value function as a critic network to estimate the expected return. In this paper, we adopt Q-actor-critic with $Q_\omega(s, a) \approx Q(s, a) = \mathbb{E}[R_t|s_t = s, a_t = a]$, where $\omega$ is the parameter set to be learned. In doing so, $Q_\omega$ (i.e., the critic) can be updated through bootstrapping and leveraged in the training of the policy network (i.e., the actor). Specifically, the loss functions for the critic and the actor are defined as follows:

$$L(\omega) = \mathbb{E}_\mathcal{D}[(\gamma Q_\omega(s_{t+1}, a_{t+1}) + r_t - Q_\omega(s_t, a_t))^2], \tag{4}$$

$$L(\theta) = \mathbb{E}_\mathcal{D}[Q_\omega(s_t, a_t) \log \pi_\theta(a_t|s_t)], \tag{5}$$

where the experience replay buffer $\mathcal{D}$ contains transitions $(s_t, a_t, r_t, s_{t+1}, a_{t+1})$, which are collected during solving a batch of instances. $\gamma Q_\omega(s_{t+1}, a_{t+1}) + r_t$ is the one-step temporal-difference (TD) target to update the critic.

The above Q-actor-critic is not directly applicable to the primitive high-dimensional action spaces in our IP problems. To learn the factorized policies we designed in Section 4.2, one way is to customize it by training an actor and critic on each dimension, and updating the parameters via the loss functions:

$$\tilde{L}(\omega) = \mathbb{E}_\mathcal{D}[\frac{1}{n}\sum_{i=1}^n (\gamma Q_\omega(s_{t+1}, a_{t+1}^i) + r_t - Q_\omega(s_t, a_t^i))^2], \tag{6}$$

$$\tilde{L}(\theta) = \mathbb{E}_\mathcal{D}[\frac{1}{n}\sum_{i=1}^n Q_\omega(s_t, a_t^i) \log(\pi_\theta(a_t^i|s_t))], \tag{7}$$

where the parameter-sharing $Q_\omega$ is used across all dimensions as the actor $\pi_\theta$, i.e., $Q_\omega(s_t, a_t^i) = Q^i(s_t, a_t^i), \forall i \in \{1, \ldots, n\}$. However, our experiments show that the critic trained by bootstrapping on each elementary action delivers inferior performance, which is similar to the finding in [26]. Intuitively, this may stem from excessive action-value regressions with one single neural network. To circumvent this issue, we keep the global TD learning in Equation (4) and adjust elementary policies by the $Q$ value of the state-action pair $(s_t, a_t)$, such that:

$$\tilde{L}(\theta) = \mathbb{E}_\mathcal{D}[Q_\omega(s_t, a_t)\sum_{i=1}^n \log(\pi_\theta(a_t^i|s_t))], \tag{8}$$

where the critic $Q_\omega$ is structured in the same manner as $\pi_\theta$, except that: 1) we add a binary value to the raw features of each variable $a_t^i$ to indicate whether it is selected; 2) we use MLP to process the graph embedding, which aggregates the embeddings of variables by mean-pooling, to output a real value that represents the $Q$ value.

**Clipping & masking.** To enhance exploration, we clip the probabilities of being selected for each variable in a range $[\epsilon, 1 - \epsilon]$, $\epsilon < 0.5$. It helps avoid always or never traversing some variables with extreme probabilities. We also do not consider empty or universal sets of variables, which lead to unsuitable sub-IPs. Though the chance to select these two sets are low, we mask them by resampling.

Details of the training algorithm are given in Appendix A.2. Besides the policy network we designed in Section 4.3, we also adopt this algorithm to train a MLP based semi-shared network similar to the one in [26], which indicates that the fully-shared one (ours) is more suitable for IP problems. More details are given in Appendix A.3.

# 5 Experimental results

We perform experiments in this section on four NP-hard benchmark problems: Set Covering (SC), Maximal Independent Set (MIS), Combinatorial Auction (CA) and Maximum Cut (MC), which are widely used in existing works. Our code is available.[3]

**Instance generation.** We generate SC instances with 1000 columns and 5000 rows following the procedure in [42]. MIS instances are generated following [43], where we use the Erdős-Rényi random graphs with 1500 nodes and set the affinity number to 4. CA instances with 2000 items and 4000 bids are generated according to arbitrary relationships in [44]. MC instances are generated according to Barabasi-Albert random graph models [45], with average degree 4, and we adopt graphs of 500 nodes. For each problem type, we generate 100, 20, 50 instances for training, validation, and testing. In addition, we double and quadruple the number of variables and generate 50 larger and even larger instances respectively for each problem, to verify the generalization performance. We name the instance groups and display their average sizes in Table 1.

Table 1: Average sizes of problem instances.

| Num. of | Training | | | | Generalization | | | | | | | |
|---|---|---|---|---|---|---|---|---|---|---|---|---|
| | SC | MIS | CA | MC | $SC_2$ | $MIS_2$ | $CA_2$ | $MC_2$ | $SC_4$ | $MIS_4$ | $CA_4$ | $MC_4$ |
| Variables | 1000 | 1500 | 4000 | 2975 | 2000 | 3000 | 8000 | 5975 | 4000 | 6000 | 16000 | 11975 |
| Constraints | 5000 | 5939 | 2674 | 4950 | 5000 | 11932 | 5357 | 9950 | 5000 | 23917 | 10699 | 19950 |

**Features.** To represent an instance, we extract static features of variables and constraints, along with the incidence matrix after presolving by the solver at the root node of B&B. In this way, redundant information could be removed and the extracted features could be more clear and compact in reflecting the problem structure. For the dynamic features, we record for each variable its value in the current solution and incumbent, as well as its average value in all incumbents to represent the solving status of LNS. Specially, at the step $t = 0$, the average incumbent values are naturally attained from the initial solution, i.e., the incumbent at the root node. Note that in practice, we concatenate the static and dynamic features for each variable, and attach them ($\mathbf{V}$) to variable nodes in the bipartite graph. The features of constraints ($\mathbf{C}$) and the incidence matrix ($A$) are attached to the constraint nodes and edges, respectively. Detailed description of the above features are available in Appendix A.4.

**Hyperparameters.** We use the state-of-the-art open source IP solver SCIP (v6.0.1) [46] as the repair operator, which also serves as a major baseline. We run all experiments on an Intel(R) Xeon(R) E5-2698 v4 2.20GHz CPU. For each problem, we train 200 iterations, during each we randomly draw $M$=10 instances. We set the training step limit $T$=50, 50, 70, 100 for SC, MIS, CA and MC respectively. The time limit for repair at each step is 2 seconds, unless stated otherwise. We use $\epsilon$=0.2 for probability clipping. For the Q-actor-critic algorithm, we set the length of the experience replay $TM$, the number of updating the network $U$=4 and the batch size $\mathcal{B}$=$TM/U$. We set the discount factor $\gamma$=0.99 for all problems, and use Adam optimizer with learning rate $1 \times 10^{-4}$. To show the applicability of our method on other solvers, we have also performed experiments where Gurobi [47] is used as the repair solver, which will be discussed in Section 5.3.

## 5.1 Comparative analysis

**Baselines.** We compare our method with four baselines:

- SCIP with default settings.

- U-LNS: a LNS version which uniformly samples a subset size, then fills it by uniformly sampling variables. We compare with it to show that our method can learn useful subset selection policies.

- R-LNS: a LNS version with hand-crafted rule proposed in [21], which randomly groups variables into disjoint equal-sized subsets and reoptimizes them in order.

- FT-LNS: the best-performing LNS version in [21], which applies forward training, an imitation learning algorithm, to mimic the best demonstrations collected from multiple R-LNS runs.

---

[3]https://github.com/WXY1427/Learn-LNS-policy

Table 2: Comparison with SCIP and LNS baselines.

| Methods | SC Obj.±Std.% | SC Gap% | MIS Obj.±Std.% | MIS Gap% | CA Obj.±Std.% | CA Gap% | MC Obj.±Std.% | MC Gap% |
|---|---|---|---|---|---|---|---|---|
| SCIP | $567.66 \pm 8.76$ | 3.62 | $-681.02 \pm 1.14$ | 0.29 | $-110181 \pm 2.03$ | 2.98 | $-852.57 \pm 1.22$ | 4.37 |
| SCIP* | $552.82 \pm 8.69$ | 0.91 | $-681.76 \pm 1.06$ | 0.18 | $-111511 \pm 1.85$ | 1.85 | $-861.10 \pm 1.26$ | 3.41 |
| SCIP** | $\mathbf{550.68 \pm 8.60}$ | **0.53** | $-682.46 \pm 1.02$ | 0.07 | $-112638 \pm 1.68$ | 0.82 | $-863.63 \pm 1.32$ | 3.13 |
| U-LNS | $568.60 \pm 12.17$ | 5.99 | $-681.38 \pm 0.95$ | 0.23 | $-103717 \pm 1.92$ | 8.67 | $-869.20 \pm 1.53$ | 2.50 |
| R-LNS | $560.54 \pm 8.07$ | 2.38 | $-682.20 \pm 0.93$ | 0.11 | $-109550 \pm 1.62$ | 3.44 | $-882.18 \pm 1.27$ | 1.05 |
| FT-LNS | $564.00 \pm 8.03$ | 3.02 | $-681.82 \pm 0.93$ | 0.17 | $-107370 \pm 2.03$ | 5.45 | $-867.05 \pm 1.64$ | 2.75 |
| Ours | $551.50 \pm 8.59$ | 0.68 | $\mathbf{-682.52 \pm 0.98}$ | **0.06** | $\mathbf{-112666 \pm 1.72}$ | **0.77** | $\mathbf{-889.61 \pm 1.32}$ | **0.27** |

[1] * and ** mean the method run with 500s and 1000s.

Table 3: Generalization to large instances.

| Methods | SC$_2$ Obj.±Std.% | SC$_2$ Gap% | MIS$_2$ Obj.±Std.% | MIS$_2$ Gap% | CA$_2$ Obj.±Std.% | CA$_2$ Gap% | MC$_2$ Obj.±Std.% | MC$_2$ Gap% |
|---|---|---|---|---|---|---|---|---|
| SCIP | $303.18 \pm 8.62$ | 6.80 | $-1323.90 \pm 0.81$ | 3.22 | $-205542 \pm 2.87$ | 7.14 | $-1691.48 \pm 1.18$ | 6.22 |
| SCIP* | $298.12 \pm 8.08$ | 5.04 | $-1357.04 \pm 1.34$ | 0.80 | $-214654 \pm 1.44$ | 3.02 | $-1706.45 \pm 1.26$ | 5.39 |
| SCIP** | $\mathbf{295.70 \pm 7.89}$ | **4.21** | $-1361.98 \pm 1.06$ | 0.44 | $\mathbf{-217271 \pm 1.93}$ | **1.84** | $-1714.71 \pm 1.02$ | 4.93 |
| U-LNS | $303.36 \pm 8.24$ | 6.90 | $-1364.66 \pm 0.69$ | 0.24 | $-197453 \pm 1.86$ | 10.79 | $-1769.00 \pm 1.03$ | 1.92 |
| R-LNS | $300.84 \pm 7.74$ | 6.06 | $-1339.00 \pm 0.81$ | 2.12 | $-204145 \pm 1.57$ | 7.77 | $-1767.09 \pm 1.00$ | 2.03 |
| FT-LNS | $303.52 \pm 7.86$ | 6.98 | $-1345.58 \pm 0.86$ | 1.63 | $-212264 \pm 1.35$ | 4.10 | $-1700.58 \pm 1.64$ | 5.72 |
| Ours | $297.90 \pm 8.20$ | 4.98 | $\mathbf{-1367.78 \pm 0.68}$ | **0.01** | $-216006 \pm 1.15$ | 2.40 | $\mathbf{-1803.71 \pm 0.92}$ | **0.00** |

| Methods | SC$_4$ Obj.±Std.% | SC$_4$ Gap% | MIS$_4$ Obj.±Std.% | MIS$_4$ Gap% | CA$_4$ Obj.±Std.% | CA$_4$ Gap% | MC$_4$ Obj.±Std.% | MC$_4$ Gap% |
|---|---|---|---|---|---|---|---|---|
| SCIP | $179.88 \pm 6.35$ | 6.37 | $-2652.56 \pm 0.58$ | 3.05 | $-372291 \pm 1.22$ | 13.44 | $-3392.02 \pm 0.86$ | 5.45 |
| SCIP* | $177.44 \pm 6.64$ | 4.91 | $-2652.56 \pm 0.58$ | 3.05 | $-372291 \pm 1.22$ | 13.44 | $-3392.94 \pm 0.82$ | 5.43 |
| SCIP** | $\mathbf{175.38 \pm 6.98}$ | **3.68** | $-2673.64 \pm 1.54$ | 2.28 | $-372291 \pm 1.22$ | 13.44 | $-3394.42 \pm 0.81$ | 5.39 |
| U-LNS | $196.60 \pm 10.13$ | 16.25 | $-2653.42 \pm 0.63$ | 3.02 | $-419973 \pm 1.11$ | 1.41 | $-3522.67 \pm 0.80$ | 1.81 |
| R-LNS | $188.02 \pm 7.13$ | 11.24 | $-2683.30 \pm 0.59$ | 1.93 | $-427478 \pm 0.95$ | 0.61 | $-3521.90 \pm 0.82$ | 1.83 |
| FT-LNS | $179.40 \pm 8.47$ | 6.04 | $-2684.94 \pm 0.81$ | 1.87 | $-424052 \pm 0.96$ | 1.41 | $-3525.28 \pm 0.83$ | 1.74 |
| Ours | $176.84 \pm 7.42$ | 4.57 | $\mathbf{-2735.86 \pm 0.50}$ | **0.00** | $\mathbf{-428052 \pm 0.12}$ | **0.49** | $\mathbf{-3587.72 \pm 0.76}$ | **0.00** |

Following [21], we tune the group number of R-LNS (and FT-LNS since it imitates R-LNS) from 2 to 5, and apply the best one to each problem. To train FT-LNS, we collect 10 demonstrations for each instance, and tune the step limit to 20 for SC, MIS, CA and 50 for MC, which perform the best. Same as our method, all LNS baselines also use SCIP as the repair operator with 2s time limit. To compare solution quality, we use the average objective value and standard deviation over the 50 testing instances as metrics. Also, since all problems are too large to be solved optimally, we measure the primal gap [19] to reflect the difference between the solution $\tilde{x}$ of a method to the best one $x^*$ found by all methods. We compute $|\mu^\top \tilde{x} - \mu^\top x^*|/max\{|\mu^\top \tilde{x}|, |\mu^\top x^*|\} \cdot 100\%$ for each instance, then average the gaps for all 50 ones. Below we report the results on testing instances of the same sizes as in training.

In this paper, we aim to improve an IP solver from externals to enable more efficient search of high-quality solutions in a broad range of COPs. To this end, we compare all methods for time-bounded optimization with the same 200s time limit, and further allow SCIP to run for longer time, i.e., 500s and 1000s. The results are gathered in Table 2. As shown, our method significantly outperforms all baselines on all problems with the same 200s time limit. It is notable that FT-LNS is inferior to R-LNS which yields demos for its imitation learning. The reason might be that FT-LNS only mimics the random demos of short (training) step limits, and hence lacks the ability of generalizing to longer steps. This limitation might hinder its application since in practice, IP problems are often solved in an *anytime* manner with flexible time/step limits. In contrast, our method avoids this myopic issue by RL training. In Appendix A.5, we also show that FT-LNS can outperform R-LNS with the same number of LNS steps as in training. Another key observation from Table 2 is that with longer time limits, SCIP is able to find better solutions than the three LNS baselines on SC, MIS and CA. However, our method still surpasses SCIP (500s) on all problems and SCIP (1000s) on MIS, CA and MC.

Table 4: Generalization to large instances (500s).

| Methods | $SC_2$ | | $MIS_2$ | | $CA_2$ | | $MC_2$ | |
| | Obj.±Std.% | Gap% | Obj.±Std.% | Gap% | Obj.±Std.% | Gap% | Obj.±Std.% | Gap% |
|---|---|---|---|---|---|---|---|---|
| SCIP** | 295.70 ± 7.89 | 4.48 | -1361.98 ± 1.06 | 0.56 | -217271 ± 1.93 | 1.59 | -1714.71 ± 1.02 | 5.44 |
| U-LNS | 302.94 ± 8.15 | 7.04 | -1368.58 ± 0.72 | 0.05 | -200256 ± 2.08 | 9.28 | -1777.98 ± 1.01 | 1.95 |
| R-LNS | 298.24 ± 7.43 | 5.39 | -1362.04 ± 0.71 | 0.52 | -207937 ± 1.44 | 5.81 | -1776.44 ± 1.02 | 2.04 |
| FT-LNS | 303.34 ± 7.97 | 7.18 | -1345.60 ± 0.83 | 1.76 | -213464 ± 1.21 | 3.30 | -1767.81 ± 1.04 | 2.51 |
| Ours | **295.36 ± 7.81** | **4.36** | **-1368.68 ± 0.65** | **0.04** | **-218920 ± 2.13** | **0.85** | **-1813.02 ± 0.91** | **0.02** |

| Methods | $SC_4$ | | $MIS_4$ | | $CA_4$ | | $MC_4$ | |
| | Obj.±Std.% | Gap% | Obj.±Std.% | Gap% | Obj.±Std.% | Gap% | Obj.±Std.% | Gap% |
|---|---|---|---|---|---|---|---|---|
| SCIP** | 175.38 ± 6.99 | 5.21 | -2673.64 ± 1.54 | 2.41 | -372291 ± 1.22 | 14.98 | -3394.42 ± 0.81 | 6.04 |
| U-LNS | 185.62 ± 8.19 | 11.26 | -2737.24 ± 0.54 | 0.09 | -426480 ± 0.93 | 2.60 | -3556.69 ± 0.80 | 1.55 |
| R-LNS | 172.96 ± 6.43 | 3.73 | -2736.60 ± 0.52 | 0.11 | -431786 ± 1.03 | 1.39 | -3554.98 ± 0.80 | 1.59 |
| FT-LNS | 175.20 ± 6.59 | 5.11 | -2685.30 ± 0.81 | 1.97 | -431234 ± 0.91 | 1.52 | -3526.24 ± 0.79 | 6.10 |
| Ours | **172.38 ± 7.14** | **3.36** | **-2738.24 ± 0.50** | **0.04** | **-437880 ± 0.72** | **0.00** | **-3612.52 ± 0.74** | **0.00** |

## 5.2 Generalization analysis

Training deep models that perform well on larger problems is a desirable property for solving IPs, since practical problems are often large-scale. Here we evaluate such generalization performance on instances with the average sizes listed in Table 1. We run our method and baselines on these instances with the same time limits as those in the experiments for comparative analysis. For our method and FT-LNS, we directly apply the policies trained in Section 5.1.

All results are displayed in Table 3. It is revealed that with the same 200s time limit, while the LNS baselines only outperform SCIP on specific problems, our LNS policies trained on small instances are consistently superior to all baselines, showing a stronger generalization ability. Also, our method delivers much smaller gaps, e.g., at least $38.35\%$ smaller than that of SCIP (200s), which are more prominent than the results in Section 5.1. It indicates that our policies are more efficient in improving SCIP for larger instances solely by generalization. When SCIP runs with 500s, it surpasses all three LNS baselines on $SC_2$, $CA_2$ and $SC_4$, while our method can still deliver better results on all problems. Compared to SCIP with 1000s, our method is inferior on $SC_2$, $CA_2$ and $SC_4$ but apparently better on the remaining 5 instance groups.

We further test all methods with 500s time limit except SCIP, which is allowed to run with 1000s. All results are gathered in Table 4. It is revealed that our method still has clear advantages over others on all problems, and consistently outperforms SCIP with mere 1/2 runtime. We find that LNS baselines can outperform SCIP on some problems, especially the three largest ones, i.e., $MIS_4$, $CA_4$ and $MC_4$. It suggests that as the problem size becomes larger, LNS could be more effective to deliver high-quality solutions by solving successive sub-IPs which have much lower complexity than the original problem. On the other hand, our method outperforms all LNS baselines, showing that it is more efficient in improving the solution. In summary, our LNS policies learned on small instances generalize well to larger ones, with a persistent advantage over other methods.

## 5.3 Experiments with Gurobi

Our LNS framework is generally applicable to any IP solver. Here we evaluate its performance by leveraging Gurobi (v9.0.3) [47] as the repair operator. Gurobi is a commercial solver and offers less interfaces to the internal solving process. Thus, we condense the static features of variables to mere objective coefficients, and attain the initial solution by running the solver with 2s time limit. During training, we set the step limit $T$=50 with 1s time limit for Gurobi at each step. For FT-LNS, we use the same 1s time limit, and tune the group number to 2 in all problems for its best performance. The remaining settings are the same as those with SCIP. During testing, we let all methods run with 100s time limit, roughly twice as much as that for training. To save space, we only show results of two instance groups for each problem in Table 5.[4] Despite the shorter runtime, we find that Gurobi (100s) generally attains lower objective values than SCIP (200s), showing a better performance to solve IP

---

[4]Specially, we observe that MIS is fairly easy for Gurobi (76 out of 100 instances can be solved optimally with average 40s). Thus, we evaluate this problem with less time limit and show the results in Appendix A.6.

Table 5: Results with Gurobi. The left part shows the results of inference on the testing set in SC, CA and MC; the right part shows the results of generalization to larger instances in $SC_2$, $CA_2$ and $MC_2$.

| Methods | SC | | CA | | MC | | $SC_2$ | | $CA_2$ | | $MC_2$ | |
|---|---|---|---|---|---|---|---|---|---|---|---|---|
| | Obj.±Std.% | Gap% | Obj.±Std.% | Gap% | Obj.±Std.% | Gap% | Obj.±Std.% | Gap% | Obj.±Std.% | Gap% | Obj.±Std.% | Gap% |
| Gurobi | 554.94 ± 8.34 | 1.15 | -111668 ± 1.96 | 1.10 | -863.91 ± 3.77 | 3.31 | 302.52 ± 7.73 | 2.43 | -214271 ± 1.52 | 3.63 | -1652.83 ± 3.63 | 5.81 |
| U-LNS | 562.08 ± 8.16 | 2.46 | -110402 ± 1.67 | 2.21 | -862.59 ± 1.75 | 3.45 | 301.48 ± 7.62 | 2.07 | -218986 ± 1.42 | 1.51 | -1733.57 ± 1.22 | 1.21 |
| R-LNS | 563.98 ± 8.29 | 2.81 | -110230 ± 1.56 | 2.36 | -860.22 ± 1.96 | 3.72 | 302.86 ± 7.41 | 2.56 | -219462 ± 1.16 | 1.29 | -1723.75 ± 1.32 | 1.77 |
| FT-LNS | 564.14 ± 8.37 | 2.84 | -110041 ± 1.56 | 2.53 | -866.22 ± 1.65 | 3.03 | 348.50 ± 9.05 | 17.99 | -206189 ± 1.39 | 7.26 | -1726.07 ± 1.19 | 1.64 |
| Ours | **551.88 ± 8.31** | **0.59** | **-111787 ± 2.60** | **1.00** | **-888.97 ± 1.55** | **0.50** | **297.70 ± 7.40** | **0.80** | **-222346 ± 1.35** | **0.00** | **-1752.98 ± 1.21** | **0.11** |

problems. Hence, the LNS baselines lose their advantages over the solver on several problems, e.g., SC, CA and $SC_2$. In contrast, our method outperforms Gurobi across all problems, showing good performance on instances of both training and generalization sizes. Moreover, our method can be well applied to much larger instances and we provide this evaluation in Appendix A.7.

## 5.4 Testing on MIPLIB

The mixed integer programming library (MIPLIB) [48] contains real-world COPs from various domains. Since the instances in MIPLIB are severely diverse in problem types, structures and sizes, it is not a very suitable testing set to directly apply learning based models and thus seldom used in the related works. We evaluate our method (with Gurobi as the repair solver) on this realistic dataset, in the style of active search on each instance [5, 6], and compare it to SCIP and Gurobi. Results show that: 1) with the same 1000s time limit, our method is superior to both solvers on 24/35 instances and comparable to them on 9/35 instances; 2) our method with 1000s time limit outperforms both solvers with 3600s time limit on 13/35 instances; 3) for an open instance, we find a better solution than the best known one. More details are provided in Appendix A.8.

## 6 Conclusions and future work

We propose a deep RL method to learn LNS policy for solving IP problems in bounded time. To tackle the issue of large action space, we apply action factorization to represent all potential variable subsets. On top of it, we design a parameter-sharing GNN to learn policies for each variable, and train it by a customized actor-critic algorithm. Results show that our method outperforms SCIP with much less time, and significantly surpasses LNS baselines with the same time. The learned policies also generalize well to larger problems. Furthermore, the evaluation of our method with Gurobi reveals that it can effectively improve this leading commercial solver. For limitations, since we mainly aim to refine off-the-shelf solvers for general IP problems, it is not sufficient to conclude that our method can transcend specialized and highly-optimized algorithms in different domains. In a practical view, our method could be a choice when new IP problems are produced with little expertise, or extensive dependence on domain knowledge is expected to be avoided. Also, our LNS policies are more suitable to improve solvers for large-scale problems in bounded time, but cannot provide optimality guarantee. For future work, we will apply our method to other (mixed) IP problems, and extend it by combining with other learning techniques for IPs, such as learning to branch.

## Acknowledgments

This research was conducted at Singtel Cognitive and Artificial Intelligence Lab for Enterprises (SCALE@NTU), which is a collaboration between Singapore Telecommunications Limited (Singtel) and Nanyang Technological University (NTU) that is supported by A*STAR under its Industry Alignment Fund (LOA Award number: I1701E0013). Wen Song was supported by the National Natural Science Foundation of China under Grant 62102228, and the Young Scholar Future Plan of Shandong University under Grant 62420089964188. Zhiguang Cao was supported by the National Natural Science Foundation of China under Grant 61803104.

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
