# Learning Large Neighborhood Search Policy for Integer Programming (Appendix)

## A.1 Architecture of bipartite GCN

In this paper, we propose to factorize the selection of a variable subset into decisions on selection of each variable, under our LNS framework. To represent such action factorization, we employ the bipartite GCN as the destroy operator, as shown in Figure A.1. In specific, the bipartite GCN comprises two stacks of graph convolution layers to compute the embeddings of variables, and one MLP module that computes probabilities of selecting each variable in parallel.

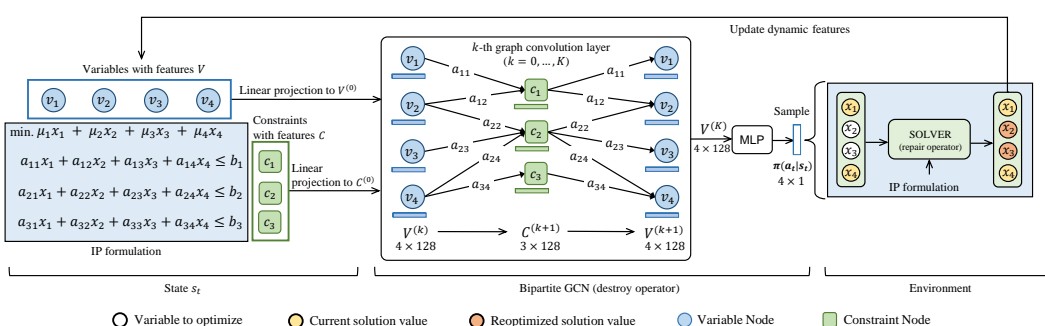

Figure A.1: Illustration of our LNS framework with the bipartite GCN based destroy operator.

## A.2 Training details

Our RL algorithm for training LNS policies is depicted by the pseudo code in Algorithm 1. Compared to the standard actor-critic algorithm, we use experience replay to empower the reuse of past samples (lines 2-8). In addition, we customize the standard Q-actor-critic algorithm for the proposed action factorization, by specializing the loss functions.

---

**Algorithm 1:** Customized Q-actor-critic for LNS

---

**Input:** actor $\pi_\theta$ with parameters $\theta$; critic $Q_\omega$ with parameters $\omega$; empty reply buffer $\mathcal{D}$; number of iterations $J$; step limit $T$; number of updates $U$; learning rates $\alpha_\theta, \alpha_\omega$; discount factor $\gamma$.

1   **for** $j = 1, 2, \cdots, J$ **do**
2     draw $M$ training instances;
3     **for** $m = 1, 2, \cdots, M$ **do**
4       **for** $t = 1, 2, \cdots, T$ **do**
5         sample $a_t^i \sim \pi_\theta(a_t^i|s_t)$, derive the union $a_t$ ;
6         receive reward $r_t$ and next state $s_{t+1}$;
7         sample $a_{t+1}^i \sim \pi_\theta(a_{t+1}^i|s_{t+1})$, derive $a_{t+1}$;
8         store transition $(s_t, a_t, r_t, s_{t+1}, a_{t+1})$ in $\mathcal{D}$;

9     **for** $u = 1, 2, \cdots, U$ **do**
10       randomly sample a batch of transitions $\mathcal{B}$ from $\mathcal{D}$;
11       update the parameters of actor and critic with $y_t = \gamma Q_\omega(s_{t+1}, a_{t+1}) + r_t$;
          $\omega \leftarrow \omega + \alpha_\omega \nabla_\omega \frac{1}{|\mathcal{B}|} \sum_\mathcal{B} (y_t - Q_\omega(s_t, a_t))^2$; $z_t = \sum_{i=1}^n \log(\pi_\theta(a_t^i|s_t))$;
          $\theta \leftarrow \theta + \alpha_\theta \nabla_\theta \frac{1}{|\mathcal{B}|} \sum_\mathcal{B} Q_\omega(s_t, a_t) z_t$;

---

## A.3 Semi-shared vs. fully-shared network

As stated in the main paper, our method can be employed to train any kind of policy network that is able to represent the factorized action. We choose the fully-shared neural network since it can be generalized to different problem sizes, which is critical to solve IP problems. Nevertheless, we also experiment with a semi-shared neural network to show its performance on problem instances of the fixed size. The architecture of the neural network is displayed in the upper half of Figure A.2, which is similar to the one used for DQN based RL algorithms in [26]. Specifically, given a collection of features for each variable, we first process them by a MLP to obtain variable embeddings. Then, these embeddings are averaged and advanced by another MLP to obtain the state embedding. Lastly, we concatenate each variable embedding with the state embedding and pass them through $n$ MLPs separately to get probabilities of $n$ variables. For the fully-shared counterpart, we only replace the $n$ MLPs by a parameter-sharing MLP, as shown in the lower half of Figure A.2. All layers in MLPs are activated by Tanh except the final output by Sigmoid.

Following the experimental setting for SCIP in the main paper, we train both the fully-shared and semi-shared networks on SC, MIS, CA and MC, with the customized actor-critic algorithm we designed. We evaluate the average objective value over the validation set after each training iteration. All training curves of initial 50 iterations are displayed in Figure A.3. We find that the fully-shared network is able to learn efficiently on SC, CA and MC, while semi-shared network performs better on MIS. It indicates that our training algorithm can be applied to different kinds of policy networks, and the fully-shared network is more effective in learning LNS policies for IP problems. For the semi-shared network, despite the good performance with relatively low-dimensional action spaces in [26], it needs far more sub-networks in our RL tasks with thousands of action dimensions, which are intractable to train together and also prevent the generalization to different problem sizes.

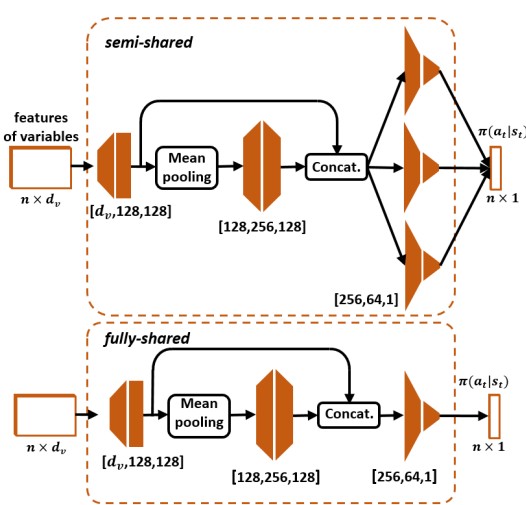

Figure A.2: Architectures of the semi-shared and fully-shared policy networks.

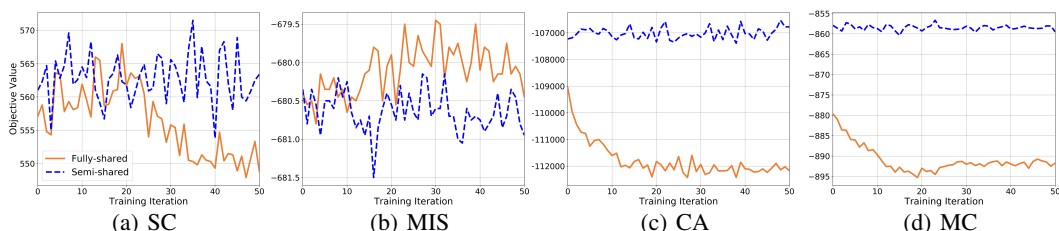

|  (a) SC | (b) MIS | (c) CA | (d) MC |

Figure A.3: Training curves of the semi-shared and fully-shared networks.

## A.4 State features

In this paper, we represent the state by a bipartite graph $\mathcal{G} = (\mathcal{V}, \mathcal{C}, \mathbf{A})$ attached by the features of variables, constraints and edges (i.e. $\mathbf{V}$, $\mathbf{C}$ and A), which are listed in Table A.1. The logic behind this design is to reflect the instance information and the dynamic solving status, both of which are critical to learn effective policies in LNS. For the static features, we consider the ones used in [20], which learns variable selection policies in B&B algorithm. It has been shown that these features have the potential to predict variables for branching. From the perspective of learning LNS, we only extract these features at the root node as the instance information. Also, we preprocess these features

in two ways: 1) we delete the ones with zero variance from the features of variables, which are the same constant across all training instances; 2) we only use the right-hand-side (RHS) vector as the features for constraints. For dynamic features, we consider the efficiency of the solving process and directly record values of the current solution and incumbent at each step of LNS. These dynamic features are linked with the static features of variables and then attached to variable nodes $\mathcal{V}$.

Table A.1: The list of features for variables, constraints and edges. *S.* and *D.* denote the static and dynamic attribute, respectively.

| Feature Types | Description | Length | *S./D.* |
|---|---|---|---|
| Variable features ($\mathbf{V}$) | Normalized reduced cost. | 1 | *S.* |
| | Normalized objective coefficient. | 1 | *S.* |
| | Normalized LP age. | 1 | *S.* |
| | Equality of solution value and lower bound, 0 or 1. | 1 | *S.* |
| | Equality of solution value and upper bound, 0 or 1. | 1 | *S.* |
| | Fractionality of solution value. | 1 | *S.* |
| | One-hot encoding of simplex basis status (i.e., lower, basic, upper). | 3 | *S.* |
| | Solution value at root node. | 1 | *S.* |
| | Solution value at the current step. | 1 | *D.* |
| | Value in the incumbent. | 1 | *D.* |
| | Average value in historical incumbents. | 1 | *D.* |
| Constraint features ($\mathbf{C}$) | Constraint right-hand side. | 1 | *S.* |
| Edge features ($A$) | Coefficient in incidence matrix. | 1 | *S.* |

## A.5   Comparison of short-term performance

In the main paper, we found that FT-LNS cannot outperform R-LNS with the long time limit due to its poor generalization to unseen states. However, according to [21], FT-LNS can outperform R-LNS with relatively short time limit, i.e. the similar runtime used in training of FT-LNS. To verify this point, we compare our method and LNS baselines with such setting. Specifically, we first test FT-LNS with the same number of LNS steps as in its training, and record its runtime. Then we test our method and other baselines using the runtime of FT-LNS as time limit. All results are summarized in Table A.2. As shown, FT-LNS can indeed surpass R-LNS on all problems, which indicates its effectiveness in short-term improvement and is consistent with [21]. Nevertheless, for these experiments with short time limits, it is clear that our method still consistently outperforms all LNS baselines with smaller gaps across all problems.

Table A.2: Results with short time limits.

| | SC | | MIS | | CA | | MC | |
|---|---|---|---|---|---|---|---|---|
| Methods | Obj.±Std.% | Gap% | Obj.±Std.% | Gap% | Obj.±Std.% | Gap% | Obj.±Std.% | Gap% |
| SCIP | 586.72 ±9.14 | 3.38 | -659.78±1.14 | 2.23 | -93715±2.64 | 8.34 | -840.68±1.49 | 3.93 |
| U-LNS | 615.78±12.68 | 12.44 | -660.30±1.13 | 2.15 | -99302±2.75 | 2.77 | -852.39±1.65 | 2.59 |
| R-LNS | 588.70±8.54 | 3.79 | -667.54±1.14 | 1.08 | -98705±1.93 | 3.37 | -851.57±1.45 | 2.68 |
| FT-LNS | 578.38±9.23 | 1.89 | -670.48±1.13 | 0.64 | -100323±2.00 | 1.89 | -865.01±1.72 | 1.15 |
| Ours | **575.80±8.94** | **1.46** | **-673.78±0.11** | **0.15** | **-100867±2.11** | **1.37** | **-872.47±1.24** | **0.30** |

## A.6   Testing on MIS with Gurobi

On the training set of MIS, Gurobi solves most instances optimally with 40s in average. As stated in the main paper, our method aims to improve solvers in bounded time and cannot guarantee optimality. Thus, in contrast to 100s time limit for other problems, we evaluate all methods on MIS with 20s. The other experimental settings follow those in Section 5.3. The results are displayed in Table A.3. As shown, our method effectively improves Gurobi to achieve smaller gaps, although it is already able to deliver high-quality solutions quickly. Also, our method is consistently superior to baselines on all instances groups, demonstrating good generalization to different-sized problems.

Table A.3: Results on MIS with Gurobi.

| Methods | MIS | | MIS$_2$ | | MIS$_4$ | |
|---|---|---|---|---|---|---|
| | Obj.±Std.% | Gap% | Obj.±Std.% | Gap% | Obj.±Std.% | Gap% |
| Gurobi | 682.22 ± 1.06 | 0.08 | -1359.54 ± 0.84 | 0.59 | -2645.88 ± 1.10 | 2.96 |
| U-LNS | 682.02 ± 0.95 | 0.11 | -1359.36 ± 0.74 | 0.61 | -2723.44 ± 0.52 | 0.12 |
| R-LNS | 681.82 ± 0.98 | 0.14 | -1365.64 ± 0.67 | 0.15 | -2722.60 ± 0.52 | 0.15 |
| FT-LNS | 682.20 ± 0.94 | 0.09 | -1358.86 ± 0.77 | 0.64 | -2722.18 ± 0.56 | 0.16 |
| Ours | **682.24 ± 0.94** | **0.08** | **-1367.48 ± 0.65** | **0.02** | **-2724.08 ± 0.52** | **0.09** |

## A.7 Generalization with Gurobi

Here we further evaluate the generalization of our LNS framework with Gurobi as the repair operator. We test all methods with 100s time limit, same as in Section 5.3. For FT-LNS and our method, the learned policies on small instances are directly used. The results are summarized in Table A.4. We can observe that while our method is slightly inferior to U-LNS and R-LNS on SC$_4$, it can still generalize well to much larger instances on the problems CA$_4$ and MC$_4$, and outperform all baselines. This indicates that our method has a good generalization ability to improve Gurobi, the leading commercial solver, for solving instances of different scales.

Table A.4: Generalization to large instances with Gurobi.

| Methods | SC$_4$ | | CA$_4$ | | MC$_4$ | |
|---|---|---|---|---|---|---|
| | Obj.±Std.% | Gap% | Obj.±Std.% | Gap% | Obj.±Std.% | Gap% |
| Gurobi | 183.60 ± 7.29 | 5.30 | -377557 ± 0.85 | 13.87 | -3373.11 ± 1.05 | 1.46 |
| U-LNS | 176.84 ± 6.89 | 1.43 | -436224 ± 1.12 | 0.49 | -3388.31 ± 0.73 | 1.02 |
| R-LNS | **176.08 ± 6.47** | **1.01** | -435669 ± 0.67 | 0.62 | -3389.47 ± 0.70 | 0.98 |
| FT-LNS | 201.14 ± 9.65 | 15.35 | -395027 ± 3.56 | 9.89 | -3373.20 ± 2.31 | 1.48 |
| Ours | 177.66 ± 6.65 | 1.92 | **-437735 ± 0.80** | **0.15** | **-3390.32 ± 0.73** | **0.95** |

## A.8 Testing on real-world instances in MIPLIB

In this appendix, we provide details of the experiment on real-world instances in MIPLIB. These instances are grouped into "easy", "hard" and "open", according to their difficulties to solve. Since our method is more suitable for large-scale IP problems, we filter out the "easy" instances with relatively small sizes. We also filter out those instances that both SCIP and Gurobi cannot find any feasible solution with 3600s time limit, and finally choose 35 representative "hard" or "open" instances with only integer variables. Among the chosen instances, the number of variables ranges from 150 to 393800 (the average is 49563), and the number of constraints range from 301 to 850513 (the average is 96778). Also, these instances cover the typical application of COP from distinct domains, e.g., vehicle routing, cryptographic research and wireless network. To cope with such heterogeneous problems, we employ our method in the active search mode. Specifically, we apply the customized Q-actor-critic in Algorithm 1 to each instance, with only two instances solved in each iteration, i.e., $M = 2$. In doing so, we can save computation memory and also raise the frequency of training. We use Gurobi as the repair operator and set its time limit to 2s in each LNS step. In addition, we set the step limit $T$=100, number of updates $U$=10, and batch size $\mathcal{B}$=32. For the initial solution, we use the one returned by Gurobi with 100s time limit. We set the whole time limit of active search to 1000s, and compare the results of SCIP and Gurobi with 1000s and 3600s time limits. The other settings follow those in Section 5.3.

All results are displayed in Table A.5. As shown, the proposed LNS framework can improve the solver effectively, and achieve better solutions than SCIP and Gurobi for most instances with the same or less runtime. Moreover, for the open instance "neos-3682128-sandon", we managed to find a new best solution.

Table A.5: Results on MIPLIB. The "BKS" column lists the best know solutions given in MIPLIB. **Bold** and * mean our method outperforms the solvers with 1000s and 3600s respectively. "-" means no feasible solution is found.

| Instance | SCIP (1000s) | SCIP (3600s) | Gurobi (1000s) | Gurobi (3600s) | Ours (1000s) | BKS |
|---|---|---|---|---|---|---|
| a2864-99blp | -71 | -71 | -72 | -72 | -72 | -257 |
| bab3 | - | - | -654709.9511 | -655388.1120 | **-654912.9204** | -656214.9542 |
| bley_xs1noM | 5227928.57 | 5227928.57 | 3999391.53 | 3938322.37 | **3975481.35** | 3874310.51 |
| cdc7-4-3-2 | -230 | -230 | -253 | -257 | **-276*** | -289 |
| comp12-2idx | - | 676 | 416 | 380 | **363*** | 291 |
| ds | 509.5625 | 461.9725 | 309 | 177 | 319 | 93.52 |
| ex1010-pi | 254 | 248 | 241 | 239 | **238*** | 235 |
| graph20-80-1rand | -1 | -1 | -3 | -6 | **-6** | -6 |
| graph40-20-1rand | -1 | -1 | 0 | -15 | **-14** | -15 |
| neos-3426085-ticino | 234 | 232 | 226 | 226 | 226 | 225 |
| neos-3594536-henty | 410578 | 410578 | 402572 | 401948 | **402426** | 401382 |
| neos-3682128-sandon | 40971070.0 | 35804070.0 | 34674767.94751 | 34666770.0 | **34666765.12338*** | 34666770 |
| ns1828997 | 43 | 32 | 145 | 133 | **128*** | 9 |
| nursesched-medium-hint03 | 8074 | 8074 | 144 | 115 | **131** | 115 |
| opm2-z12-s8 | -36275 | -38015 | -33269 | -33269 | **-53379*** | -58540 |
| pb-grow22 | 0.0 | 0.0 | -31152.0 | -46217.0 | **-46881.0*** | -342763.0 |
| proteindesign121hz512p9 | - | - | 1499 | 1499 | **1489*** | 1473 |
| queens-30 | -33 | -39 | -39 | -39 | -39 | -40 |
| ramos3 | 242 | 242 | 252 | 245 | **248** | 192 |
| rmine13 | -611.536750 | -611.536750 | -2854.351313 | -3493.781904 | **-3487.807859** | -3494.715232 |
| rmine15 | -759.289522 | -759.289522 | -192.372262 | -1979.559046 | **-5001.279118*** | -5018.006238 |
| rococoC12-010001 | 44206.0 | 38905.0 | 35463 | 34673 | **35443** | 34270 |
| s1234 | 319 | 319 | 41 | 29 | 41 | 29 |
| scpj4scip | 141 | 141 | 134 | 133 | 134 | 128 |
| scpk4 | 346 | 342 | 331 | 330 | **329*** | 321 |
| scpl4 | 296 | 296 | 281 | 279 | 281 | 262 |
| sorrell3 | -11 | -15 | -16 | -16 | -16 | -16 |
| sorrell4 | -18 | -18 | -22 | -23 | **-23** | -24 |
| sorrell7 | -152 | -152 | -183 | -187 | **-188*** | -196 |
| supportcase2 | - | - | 397 | 397 | 397 | 109137 |
| t1717 | 236546 | 228907 | 201342 | 201342 | **195894*** | 158260 |
| t1722 | 138927 | 138927 | 123984 | 117171 | **117978** | 109137 |
| tokyometro | - | 33134.6 | 8493.3 | 8479.5 | 8582.70 | 8329.4 |
| v150d30-2hopcds | 42 | 41 | 41 | 41 | 41 | 41 |
| z26 | -1029 | -1029 | -1005 | -1083 | **-1171*** | -1187 |