# OpenReview forum: "Learning Large Neighborhood Search Policy for Integer Programming"
_NeurIPS.cc/2021/Conference — NeurIPS 2021 Spotlight_

### Official Review · Reviewer_r59Z · 2021-07-13

**Rating:** 7
**Confidence:** 2

**Summary:**

This paper presents a reinforcement learning based approach for tuning a heuristic method for integer programming solvers. It derives the method, then validates its performance computationally.

**Main Review:**

The paper is at times difficult to read. I would request that the authors attempt to polish the presentation.

I cannot assess the quality of proposed RL method, or the novelty with respect to the literature on heuristic learning. Instead, I will focus on the computational experiments. Here, the results seem promising, though I have a few questions about the methodology, as well as a concern with the baseline used (detailed below).

* L41: What does "externals" mean here?
* L108: IP is _not_ a subset of COP, it is in fact much more general.
* L137: "latter" seemingly refers to the incidence matrix, which does not make sense in context.
* L139: What if you do not have an initial solution? Often for COPs and IPs producing feasible solutions is very difficult. How does your method accomodate this, if at all?
* L182: "parametrize"
* Is a single model used throughout the computational section for each set of experiments? Or are different models trained for each problem class (SC, MIS, CA, MC, MIPLIB)?
* The usage of SCIP and Gurobi with default settings is a bit of a strawman. These solvers are configured to prove optimality quickly, while LNS is aimed solely at producing primal solutions. As such, the solvers will spend significant times improving the dual bound through cuts, presolve, etc. This renders wall time comparisons unfair to the solvers. A more fair comparison would configure the solvers to focus on the primal bound, using their respective "emphasis" parameters.
* L299: It would be good to compare also against the best dual bound available to get a sense of whether the produced solutions are close to optimal. You can get these dual bounds from SCIP or Gurobi. Without such a comparison, it is difficult to gauge how much improvement is offered by the new method.
* Table 1: "Constrains"
* L349: MIPLIB contains many instances which are not COPs.
* L356: If you have produced a new best solution for a MIPLIB instance, you should submit it upstream!

**Time Spent Reviewing:**

1.5

---

> ### Author Response · Authors · 2021-08-10
> **Response to Reviewer r59Z**
>
> We appreciate the reviewer for the valuable and positive comment. According to your advice, we will improve the writing in the paper for better presentation.
>
> Our responses to your questions and concerns are listed as below:
>
> **1. L41: What does "externals" mean here?**
>
> The “externals” means our method aims to improve the performance of a solver without the interface access to its internal solving process. Most existing works attempt to learn a certain heuristic (e.g., the branching heuristic) used in the B&B tree search process, so they have to extend the interface or plugins to interact with the IP solver during the search process, e.g., to collect features of the solving status or coordinate the new heuristic with other components. In contrast, our method directly uses the solver to compute a series of sub-IPs without modifying its interiors. We only learn policies to provide proper sub-IPs to the solver so as to guide the solving process.
>
> **2. L108: IP is not a subset of COP, it is in fact much more general.**
>
> We double check relevant concepts and agree that our original description is misleading. We will modify the description accordingly.
>
> **3. L137: "latter" seemingly refers to the incidence matrix, which does not make sense in context.**
>
> The "latter" refers to the dynamic solving status in L136. We will revise the description here to make the presentation more clear.
>
> **4. L139: What if you do not have an initial solution? Often for COPs and IPs producing feasible solutions is very difficult. How does your method accomodate this, if at all?**
>
> In this paper, we mainly aim at proposing a new learning based method to quickly improve feasible solutions of IPs. However, it is interesting and valuable to investigate more challenging situations where finding feasible solutions is extremely difficult. Our method has a potential to be adapted to this scenario, by learning to find promising sub-problems that lead to feasible solutions.
>
> Intuitively, we could start with an infeasible solution and still train the policy network to select variables to be optimized at each step. In this new MDP, we could set negative rewards to penalize the actions (i.e., the selected variables) which produce infeasible solutions, according to their degrees of constraint violation. In doing so, the variables, which might deliver feasible solutions, will take higher probabilities to be selected and optimized. In addition, the action elimination techniques in RL could be used to accelerate the reduction of infeasible solution subspaces. We will investigate the above potentials in future work.
>
> **5. L182: "parametrize"**
>
> The "parametrize" is an alternative form of “parameterize” and we find it is also used in the literature. However, we will change it to avoid confusion in reading.
>
> **6. Is a single model used throughout the computational section for each set of experiments? Or are different models trained for each problem class (SC, MIS, CA, MC, MIPLIB)?**
>
> In our experiments, we train different models for each problem class. Specifically, we train one policy network for a problem class with instances in the training set, and test it on different-sized instances.
>
> **7. The usage of SCIP and Gurobi with default settings is a bit of a strawman. These solvers are configured to prove optimality quickly, while LNS is aimed solely at producing primal solutions. As such, the solvers will spend significant times improving the dual bound through cuts, presolve, etc. This renders wall time comparisons unfair to the solvers. A more fair comparison would configure the solvers to focus on the primal bound, using their respective "emphasis" parameters.**
>
> We would like to note that most existing works focus on improving a solver with its default settings, with only a certain inner policy replaced by a learning model. In our work, we highlight a high-level and learning based LNS framework to refine a solver from externals, and have demonstrated this point with both SCIP and Gurobi, without further modifications on their settings.
>
> We also notice that default settings of solvers are practically a trade-off between proving optimality and quickly finding feasible solutions. The focus on these two aspects can be tuned by setting certain parameters. Hence we conduct a quick test for comparisons to SCIP and Gurobi with tuned "emphasis" parameters, by turning on the “aggressive” mode in SCIP and setting MIPFocus=1 in Gurobi, such that they tend to search for primal solutions more quickly. With the same runtime (200s) in our paper, the results on the set covering problem are shown as follows:
>
> |              |      SC      |      SC2     |      SC4     |
> |--------------|:------------:|:------------:|:------------:|
> | SCIP         | 560.43±9.83% | 303.26±7.94% | 178.14±6.79% |
> | Ours(SCIP)   | 551.50±8.59% | 297.90±8.20% | 176.84±7.42% |
> | Gurobi       | 548.92±8.22% | 299.62±7.93% | 181.18±7.50% |
> | Ours(Gurobi) | 551.88±8.31% | 297.70±7.40% | 177.66±6.65% |
>
> We observe from the above table that our method that uses SCIP and Gurobi with default settings generally surpasses both solvers with “emphasis” configurations. We intend to complete the test on other problems and put the results in the appendix.
>
> Lastly, we would like to mention that it is also possible to use suitably configured solvers in our framework, so that our method could benefit from their better solutions at each step. We leave this potential in our future work.
>
> **8. L299: It would be good to compare also against the best dual bound available to get a sense of whether the produced solutions are close to optimal. You can get these dual bounds from SCIP or Gurobi. Without such a comparison, it is difficult to gauge how much improvement is offered by the new method.**
>
> In this paper we use the primal gap metric, since we target at promoting the solution quality in bounded time. However, to provide a clue about the optimal gap, we test the dual bounds on set covering instances in SCIP (200s) and Gurobi (100s) and compute average gaps to them. The results are displayed in the table below. We can observe that there are obvious gaps to prove optimality in the limited time, even for the small instances (i.e. SC). On the other hand, it is clear that our method with smaller gaps is a better choice than the original solvers, if we focus on searching for better solutions quickly (i.e. the target of our work).
>
> |              |   SC   |   SC2  |   SC4  |
> |--------------|:------:|:------:|:------:|
> | SCIP         | 28.79% | 31.02% | 40.83% |
> | Ours(SCIP)   | 25.22% | 28.37% | 39.03% |
> | Gurobi       | 24.52% | 28.96% | 40.73% |
> | Ours(Gurobi) | 24.29% | 26.90% | 36.22% |
>
> **9. Table 1: "Constrains"**
>
> We will correct the mentioned typo in the paper.
>
> **10. L349: MIPLIB contains many instances which are not COPs.**
>
> We will modify the corresponding description to make it more accurate.
>
> **11. L356: If you have produced a new best solution for a MIPLIB instance, you should submit it upstream.**
>
> Thanks for the advice and we will upload the new best solution.

---

> > ### Comment · Reviewer_r59Z · 2021-08-31
> > **Response**
> >
> > I thank the authors for their detailed response. I am especially appreciative of the additional computations. In light of this, I will increase my score.

---

> > > ### Author Response · Authors · 2021-09-01
> > > **Response to Reviewer r59Z**
> > >
> > > Thanks for the acknowledgement and support.

---

### Official Review · Reviewer_WFms · 2021-07-14

**Rating:** 6
**Confidence:** 4

**Summary:**

The paper proposes to augment large neighborhood search (LNS) method in integer programming solver (e.g., SCIP) with a learning component. in particular, the paper provides a MDP formulation, parameterization and RL training pipeline for the problem. The paper also shows significant empirical performance gains compared to other baseline methods.

===== updated post rebuttal ====

The authors have addressed most of my technical concerns in details. I would like to see such clarifications shown in the final draft of the paper should it be accepted. I would like to stick with my original evaluation.

**Ethical Concerns:**

No clear ethical concerns.

**Limitations And Societal Impact:**

The authors have discussed their limitations in the conclusion section.

**Main Review:**

=========== Originality ============

Though using learning for LNS has been touched on by a number of prior work, as discussed in the paper, this paper seems to be one of the first applications to show empirical progress of using RL for improving integer programming subroutines. Though idea-wise, applying RL to such problems is not extremely novel, the paper shows much solidity in overcoming the technical challenge of applying RL.

=========== Quality ============

The paper has relatively strong quality in terms of the technical challenge the proposed approach overcomes. Purely based on the comparative tables shown in Table 1-4, the new method enjoys strong empirical gains compared all other baselines (including 'static' baselines such as SCIP, as well as learned baselines).

=========== Clarity ============

Overall, the paper is clearly written. The logic is clear and technical details are spelled out rather comprehensively. However, I do have a few technical details that need clarifying from the authors.

=========== Significance ============

The paper demonstrates some empirical success in applying RL to LNS. This bears some significance to the learning-to-optimize community, both in terms of the new approach it proposes, as well as the new domain (LNS) that the method applies.

=========== Technical questions ============

I have a few tech questions:

1. From the paper, the MDP runs for T steps, where T is between 50 and 100. How is this value chosen in practice? Do we set a larger T for more difficult problems?

2. In addition to the step limit T, there is a time limit of 2s for each step (see line 295). If I understand correctly, this 2s time limit is for the computational time it takes to approximately solve the sub-problem, in order for the MDP to transition? Do the authors find the learning and MDP transition to be greatly impacted by the 2s time limit? In other words, when the time limit is larger, at each step the solver can solve the sub-problem more optimally and hence induce a different MDP transition dynamics.

3. From above, does this mean that if T=100, then the overall time constraint is 2*100=200s? What's the connection between this and the overall time limit of 200s (see line 306)?

3. Related to the above question, I am a bit confused about the relationship between the running of LNS and the overall solving procedure. SCIP by default uses a combination of LNS, B&B, cutting planes and other primal heuristics to solve a problem. Do the authors run LNS alongside other methods (e.g. B&B and cutting planes) when doing the training and evaluation? If so, then it seems that the MDP transition function should also depend on other components of the solver such as B&B, because they can also impact the statistics and hence states of the MDP. In that case, do we specify a time budget for the LNS and an overall budget for the whole solver (and hence a time limit for other components of the solver)?

4. Do training and evaluation phase have the same behavior? In other words, do we train the LNS RL agent under a 'cleaner' setup, where all other components of the solver are turned off, while at evaluation phase, we run LNS alongside other components? This is the approach taken by the cutting plane method by Tang et al, 2020 where at training time they train the RL agent on a pure cutting plane problem, while at evaluation phase they combine the agent with other methods to jointly solve problems.


5. Is the evaluation pipeline different from how LNS is typically used in a full IP solver? I think SCIP documentation says that primal heuristics such as LNS are run every once in a while, to plunge deep into the search tree and obtain good primal solutions. On the other hand, in the paper, the evaluation seems to rely mainly on primal heuristics, i.e., the LNS is run fairly frequently. Can the authors comment on this?

6. How are validation instances used exactly? Do you run RL training on the training instances, and early stop based on performance metrics on the validation instances?

7. Comparing results for the MC instances in Table 2 and Table 3, it seems that the RL agent achieves even 'better' performance on large instances measured by the gap. Table 3 shows that RL achieves 0% gap while on the training instances in Table 2, the gap is nonzero. Is this a surprising result? RL achieves zero gaps on large instances, is it because RL finds the best solution among all other baselines, and the gap metric as defined in the paper is effectively zero?

**Time Spent Reviewing:**

3

---

> ### Author Response · Authors · 2021-08-10
> **Response to Reviewer WFms**
>
> We appreciate the reviewer for the valuable and positive comments. We list our answers to the concerned questions as follows:
>
> **1. From the paper, the MDP runs for T steps, where T is between 50 and 100. How is this value chosen in practice? Do we set a larger T for more difficult problems?**
>
> The step limit T has an impact on the performance of the learned policy. Normally, we need T to be large enough to collect sufficient states for learning. However, as the solution quality is boosted along the steps, the rewards will become sparse with a higher occurrence frequency of 0, which is natural for improvement heuristics such as LNS. The sparse reward could make the RL training more difficult.
>
> Empirically, we first run 10 random instances with the initialized (untrained) network and determine a suitable step limit with dense rewards. In other words, the training step limit T is not determined according to the difficulty of problems. Instead, we focus on picking a suitable T with more (non zero) reward signals.
>
> **2. In addition to the step limit T, there is a time limit of 2s for each step (see line 295). If I understand correctly, this 2s time limit is for the computational time it takes to approximately solve the sub-problem, in order for the MDP to transition? Do the authors find the learning and MDP transition to be greatly impacted by the 2s time limit? In other words, when the time limit is larger, at each step the solver can solve the sub-problem more optimally and hence induce a different MDP transition dynamics.**
>
> Yes. 2s time limit is used for the computation at each step to solve the sub-problem. Indeed, the choice of this time limit will influence the MDP dynamics and thus training of the policy network. In specific, it impacts the dynamic status of the states and also the rewards along the MDP, since it determines the solution returned at each step. In practice, we have tried certain time limits (i.e. from 1s to 5s) and choose 2s since it performs generally well on different problems.
>
> **3. From above, does this mean that if T=100, then the overall time constraint is 2*100=200s? What's the connection between this and the overall time limit of 200s (see line 306)?**
>
> We would like to note that the 2s time limit at each step might not be used up, if the sub-problem is very small with very few variables to optimize. So if T=100 with 2s time limit, the overall time may still be less than 200s.
>
> In our experiments, we only use the step limit T as the stop criterion in training. For testing, we let all methods run until reaching the 200s time limit for fair comparison, instead of controlling the step limit.
>
> According to the above statement, although we may use less overall time limit for some instances in training, the learned policy network has a generalization ability to keep improving solutions with longer runtime in testing.
>
> **4. Related to the above question, I am a bit confused about the relationship between the running of LNS and the overall solving procedure. SCIP by default uses a combination of LNS, B&B, cutting planes and other primal heuristics to solve a problem. Do the authors run LNS alongside other methods (e.g. B&B and cutting planes) when doing the training and evaluation? If so, then it seems that the MDP transition function should also depend on other components of the solver such as B&B, because they can also impact the statistics and hence states of the MDP. In that case, do we specify a time budget for the LNS and an overall budget for the whole solver (and hence a time limit for other components of the solver)?**
>
> In this paper, we use SCIP with default settings as the repair operator. We do not modify the inner logics and components of SCIP, such as B&B, cutting planes, primal heuristics (e.g., LNS) and so on. Instead, our learning based LNS is developed from externals of a solver. At each step, we set a time limit for the entire solver to compute the sub-problem, and attain a solution. In other words, we regard the solver as a black box to just output the solution, and use the neural network to pick variables to be optimized after the solution is returned. The solver (including its components) impacts the MDP transition by its returned solutions at each step.
>
> In our testing, the overall time limit is set for the whole solving process, including the runtime of the solver at each step, MDP transitions, etc. Please note that the runtime of the solver already includes the time of running its components, e.g., B&B, cutting planes, primal heuristics, etc.
>
> **5. Do training and evaluation phase have the same behavior? In other words, do we train the LNS RL agent under a 'cleaner' setup, where all other components of the solver are turned off, while at evaluation phase, we run LNS alongside other components? This is the approach taken by the cutting plane method by Tang et al, 2020 where at training time they train the RL agent on a pure cutting plane problem, while at evaluation phase they combine the agent with other methods to jointly solve problems.**
>
> We leave all solver components unchanged as their default settings in both training and testing. The method in Tang et al, 2020 aims to learn a policy for a specific component in the solver, that is, cutting planes. They adjust other components to highlight the effect of learning to cut. In contrast, our method uses the solver directly to solve sub-problems, without extra modifications on inner components. We focus more on learning to select variables at each step of LNS, so as to construct proper sub-problems. Or in other words, the RL agent decides what (or which) sub-problem should be fed to the solver at each step of LNS.
>
> **6. Is the evaluation pipeline different from how LNS is typically used in a full IP solver? I think SCIP documentation says that primal heuristics such as LNS are run every once in a while, to plunge deep into the search tree and obtain good primal solutions. On the other hand, in the paper, the evaluation seems to rely mainly on primal heuristics, i.e., the LNS is run fairly frequently. Can the authors comment on this?**
>
> In contrast to primal heuristics embedded in the solver and specialized for B&B, we propose a high-level LNS method, with the solver directly used as repair operator and the policy network as the destroy operator to construct sub-problems. In other words, we focus on guiding a solver by learning to provide proper sub-problems from externals, instead of designing a primal heuristic inside the solver. The merits are as follows: 1) we circumvent hand-crafted work to design sophisticated heuristics in the solver, which need much expertise and complex interface; 2) our method learns problem-specific LNS policies which have a potential to deliver better solutions, rather than general-purpose primal heuristics in a solver; 3) even from the perspective of learning, our method is more convenient for development compared to most of the previous methods, since we need less modifications and interfaces of the solver.
>
> **7. How are validation instances used exactly? Do you run RL training on the training instances, and early stop based on performance metrics on the validation instances?**
>
> In the training, we save the best-performing model on the validation set, based on the average objective value over the instances.
>
> **8. Comparing results for the MC instances in Table 2 and Table 3, it seems that the RL agent achieves even 'better' performance on large instances measured by the gap. Table 3 shows that RL achieves 0% gap while on the training instances in Table 2, the gap is nonzero. Is this a surprising result? RL achieves zero gaps on large instances, is it because RL finds the best solution among all other baselines, and the gap metric as defined in the paper is effectively zero?**
>
> In this paper, we use the primal gap as the metric, which is the relative difference between the solution found by a method and the best solution found by all methods (defined in line 297-299). We report the average primal gap over all instances in the testing set. Thus, the average primal gap of a method can achieve exactly zero if it attains best solutions for all the tested instances. On the other hand, the gap of a method could be large, if its solutions are relatively poor (e.g. the objectives are obviously larger than other methods in minimization problems).
>
> According to the above statement, the nonzero but smallest gap in Table 2 means our method can generally achieve the best or near-best solutions for most tested instances. The 0.00% gap in Table 3 means our method achieves the best or near-best solutions for all instances (Please note that the gap is not exactly 0 since more non-zero decimals are not shown).
>
> Lastly, it is normal that our method has smaller gaps on some large problems (e.g. MC), meaning that its solutions are the best or near the best on more instances compared to the results on small problems.

---

### Official Review · Reviewer_Tvg1 · 2021-07-16

**Rating:** 8
**Confidence:** 4

**Summary:**

The last few years have seen numerous projects leveraging ML/RL to
speedup the optimization of ILP problems. Many of these attempts fall
under 2 categories:
  * RL driven branch-and-bound. By integrating a policy deeply inside
an ILP solver to drive the branch&bound process, these approaches have
the ability to extract a rich set of features from the solver and can
leverage them to inform the decisions made by the policy. The work of
Gasse et al. (Exact combinatorial optimization with graph convolution
neural networks)  is an example of such an approach.
* Large Neighborhood Search. This technique has been applied with
success to speedup the optimization of integer programs, in particular
in the work of Song et al. (A general large neighborhood search
framework for solving integer linear programs). This approach is
easier to implement since it doesn't require a deep integration into
an ILP solver (it merely calls one to solve subproblems).

This paper proposes a hybrid of the two approaches. It leverages a
neural network architecture similar to the Gasse GCN as well as the
featurization introduced by Gasse to build and inform a policy that
drives the large neighborhood search pioneered by Song et al. It also
offers some novel aspects, including a customized actor critic
algorithm to train the policy.

**Limitations And Societal Impact:**

The paper is candid about the limitations of the work: it lists them in section 6 and provides some ideas to overcome them in follow up work.

**Main Review:**

The paper is well written and easy to follow. It is organized clearly, and makes it easy for the reader to follow the approach taken, and provides all the relevant technical information needed to reproduce the results. It also cites all the relevant related work.

The work makes 3 novel contributions:
* It bridges the gaps between two approaches to the field of RL driven ILP solver that have remained on parallel paths until now. The first approach modifies the branch&bound algorithm of existing ILP solvers and drives it using a trained policy. It requires a deep integration with the ILP solver, which is complex to implement but enables the extraction of a rich set of input features to inform the policy. The second approach relies on the large neighborhood search approach to generate small subproblems that can be solved quickly by an existing ILP solver. The ILP solver is used as a black box the solver which makes the implementation easier. This is an interesting idea since it has the potential to bring the best of both worlds to the table. I would be surprise if this didn't inspire other researchers to follow up in this direction.
* Inspired by previous work, it decomposes the actions into sub-actions (one per variable of the ILP) which are taken individually. The sub-actions are taken by individual policies which share parameters. The number of decisions the combined policy only needs to grow linearly with the number of variables and constraints of the ILP under consideration even through the action space grows exponentially in the size of the ILP. This is significant for two reasons: (i) and unlike previous work on ILP, this paper can leverage the entire action space and (ii) As far as I know, this is the first time that an action factorization scheme is successfully attempted on such a large action space, and could have numerous applications beyond ILP.
* It introduces a novel variation of the actor critic algorithm to train the policy.

The paper appears technically sound, and the approach is extensively compared against the SCIP solver
and the results of Song et al. The authors demonstrate that the combination
of (i) their actor critic based training (ii) a richer set of features
and (iii) the GCN based architecture outperforms Song et al and the
SCIP solver. I would have liked to see an evaluation against Gasse et
al. as well as an ablation study to get a sense of the relative
importance of these 3 factors. This might also help figure out whether
the approach of Gasse or that of Song is the most promising overall.

The previous experiments were repeated, but this time using the Gurobi
solver. Again the hybrid approach came up on top. Since the extended
set of features provided by the SCIP solver are not available when
using the Gurobi solver, it seems that the richer featurization made
available by SCIP is not critical to the outcome. This is somewhat
counter intuitive, so any insights you have about why that is would
enrich the paper.

Last but not least, RL based approaches can be sensitive to initial
conditions. I'd love to see how much variance you get when
initializing your policy with different random weights.

Overall, this work makes solid albeit incremental progress in each of the 3 directions outlined above. If the authors can address my concerns I would gladly increase the score of my review.


**Time Spent Reviewing:**

7

---

> ### Author Response · Authors · 2021-08-10
> **Response to Reviewer Tvg1**
>
> We appreciate the reviewer for the valuable comment and the strong support. Our responses to your concerns are listed as below:
>
> **1. The authors demonstrate that the combination of (i) their actor critic based training (ii) a richer set of features and (iii) the GCN based architecture outperforms Song et al and the SCIP solver. I would have liked to see an evaluation against Gasse et al. as well as an ablation study to get a sense of the relative importance of these 3 factors. This might also help figure out whether the approach of Gasse or that of Song is the most promising overall.**
>
> Due to the limited time, we have conducted a quick experiment to compare with the method in Gasse et al. [17] and the ablation study on the Set Covering problem. Specifically, we collect 10000 samples from the same-sized instances as used in our training set to train the model in [17], and evaluate it on testing sets in our paper, i.e., SC, SC2 and SC4.
>
> With the same runtime of 200s, all results are displayed in the table below. Ours (A2C) means we only use the coefficients in the formulation as features and structure the policy network using pure MLPs. Ours (A2C+GCN) means we replace the MLP with GCN. Finally, Ours (A2C+feats.+GCN) further adds the richer features used in our paper.
>
> |                      |      SC      |      SC2     |      SC4     |
> |----------------------|:------------:|:------------:|:------------:|
> | Method in [17]       | 565.52±8.44% | 302.54±7.96% | 178.64±6.19% |
> | Ours(A2C)            | 563.12±8.93% | 302.64±7.93% | 177.24±6.96% |
> | Ours(A2C+GCN)        | 552.96±8.17% | 298.16±7.76% | 177.04±7.22% |
> | Ours(A2C+feats.+GCN) | 551.50±8.59% | 297.90±8.20% | 176.84±7.42% |
>
> From the above table, we can observe that our method outperforms the method in [17] on different-sized instances. Meanwhile, the ablation study shows the effects of the three factors in our method, since each one of them could generally bring a boost to the solution quality. We will extend the above results and discussions, and add them to our paper.
>
> **2. The previous experiments were repeated, but this time using the Gurobi solver. Again the hybrid approach came up on top. Since the extended set of features provided by the SCIP solver are not available when using the Gurobi solver, it seems that the richer featurization made available by SCIP is not critical to the outcome. This is somewhat counter intuitive, so any insights you have about why that is would enrich the paper.**
>
> The experiments with Gurobi mainly aim to demonstrate the versatility of our framework. We would like to note that although the solver offers less interfaces to get more attributes of instances, we can still improve its performance by the simple featurization. This evaluation is valuable in a practical sense, considering that some commonly used solvers have limited internal access, e.g., Gurobi and Cplex. The point is, it is definitely beneficial to leverage richer features if they are easy to obtain (e.g., in SCIP), as shown in the above table. Nevertheless, our method is still effective to improve a solver with relatively limited access.
>
> **3. Last but not least, RL based approaches can be sensitive to initial conditions. I'd love to see how much variance you get when initializing your policy with different random weights.**
>
> In this paper, we use the default initializer in Tensorflow (i.e. glorot uniform initializer)  to initialize the neural networks. Here we run a quick experiment with three initializers in Tensorflow with their default settings, i.e., orthogonal initializer, random uniform initializer and truncated normal initializer, and we keep other hyper-parameters the same as in our paper. The results are shown in the following table, where “Default” means the default initializer used in our paper. It is observed that the impact of the weight initialization is relatively small. The average standard deviation of the initializers over instances in SC, SC2 and SC4 is 1.31%, 2.57% and 3.28%, respectively.
>
> We also would like to explore a suitable initialization for our method and leave this promising direction as the future work.
>
> | Initializer    |      SC      |      SC2     |      SC4     |
> |----------------|:------------:|:------------:|:------------:|
> | Orthogonal     | 559.14±8.28% | 299.22±7.37% | 174.30±7.00% |
> | Random uniform | 562.66±7.89% | 301.44±7.44% | 174.72±6.34% |
> | Truncated normal  | 553.46±8.51% | 299.08±7.98% | 176.30±7.54% |
> | Default        | 551.50±8.59% | 297.90±8.20% | 176.84±7.42% |

---

### Official Review · Reviewer_WJkM · 2021-07-17

**Rating:** 7
**Confidence:** 4

**Summary:**

In this paper, the authors propose an actor-critic reinforcement learning algorithm to learn policies for large neighborhood search (LNS). A central innovation is action factorization to tackle the high-dimensional action space. Extensive empirical evaluations are provided, covering different solver integrations (SCIP and Gurobi) as well as generalizations to larger problem instances.


**Limitations And Societal Impact:**

Adequately addressed.

**Main Review:**

Post-rebuttal comments: I thank the authors for providing detailed answers to my questions. I will keep my score.

Originality: The main idea is straightforward given the MDP formulation of LNS. Two innovations come in the form of defining factorized action spaces and the model architecture design to reflect the factorization. They are relatively small novelties but the empirical results show their efficacy.

This recent workshop paper [1] also applies RL to LNS. I think the authors should add it to the related discussion.

[1]: Neural Large Neighborhood Search, Ravichandra Addanki, Vinod Nair, Mohammad Alizadeh, https://openreview.net/forum?id=xEQhKANoVW

Quality: For an empirical paper, the authors did a good job of providing thorough analyses in the following aspects.
1. The problem instances are diverse and contain real-world problems in MIPLIB.
2. Two different solvers, SCIP and Gurobi, are used to show the generality of the proposed method.
3. I particularly like the generalization analysis. Integer programming solving typically needs good anytime performance, thus models able to generalize to larger instances are needed, which is a weakness of the previous imitation learning approach.

I have some questions regarding some details of the experiments.
1. On line 276, different step limits are used to train policies for different problems. On line 304, it is mentioned that a 200s time limit is used. Given the time limit for repair is 2 seconds. Does that mean, for example for SC problems, the solving terminates before reaching the 200s time limit?
2. Does the time limit include all runtime, such as model inference, or only include solver invocation time?
3. How many random seeds did you use for the RL results?

Clarity: This paper is clear and easy to read. Good job!

Significance: As the learning for LNS starts getting popular, this paper is a nice addition focusing on applying RL. It is relevant to the learning-to-optimize community at NeurIPS. It is likely that future works will build on it.


**Time Spent Reviewing:**

2

---

> ### Author Response · Authors · 2021-08-10
> **Response to Reviewer WJkM**
>
> We appreciate the reviewer for the valuable and positive comment.
>
> Regarding “Neural Large Neighborhood Search” [1], we find it learns a destroy policy to sequentially pick variables at each step. Thus it may constrain its application to relatively small instances, as shown in its experiments of the original paper. We will include this reference and the corresponding discussions in the related work section.
>
> Regarding the experimental details, our responses are listed as follows:
>
> **1. On line 276, different step limits are used to train policies for different problems. On line 304, it is mentioned that a 200s time limit is used. Given the time limit for repair is 2 seconds. Does that mean, for example for SC problems, the solving terminates before reaching the 200s time limit?**
>
> For training (as described in line 276), we use different step limits (50, 50, 70, 100) for different problems, with the 2 seconds repair time. So the solving process generally stops before reaching 200s. But in testing (as described in line 304), we run all methods until reaching the 200s time limit to solve each instance, instead of controlling termination using step limit.
>
> We would like to note that using fewer steps in training is more suitable for RL, because longer improvement sequences could suffer from sparse rewards. In terms of testing, we use the same runtime for different methods for fair comparison.
>
> **2. Does the time limit include all runtime, such as model inference, or only include solver invocation time?**
>
> The time limit includes all runtime of the solving process, e.g., the time for model inference, solver invocation, computation of initial solutions, etc.
>
> **3. How many random seeds did you use for the RL results?**
>
> In our experiments, we use one random seed for RL results since we find our method is fairly robust to different seeds. Specifically, we perform experiments on SC, SC2 and SC4 using five random seeds. The corresponding average standard deviations are 0.66%, 0.91% and 1.01% respectively, meaning that our results are fairly stable with varying seeds on different-sized problems.

---

### Author Response · Authors · 2021-08-10
**Thanks to all reviewers for their valuable time and efforts.**

We greatly appreciate the comments from the reviewers. In general, all feedback acknowledges the novelty of this paper and its significance in the learning-to-optimize community, in terms of both techniques and  empirical results.

We have provided detailed point-to-point responses to individual reviewers below. For your reading convenience, the reviewers’ comments are shown in Bold and listed by numbers, with each followed by our response.

---

### Decision · Program_Chairs · 2021-09-27

**Decision:**

Accept (Spotlight)

**Comment:**

All reviewers agree that this paper applies RL (Q-actor-critic method) to learn Large Neighborhood Search policy for integer linear programming (ILP) and achieves strong performance compared to default policies in popular ILP solvers (SCIP and Gurobi) in multiple domains (SC, MIS, CA and MC) that have been widely used in previous works. Some of its proposed algorithm is novel (e.g., training an independent policy for each variable, new network architecture design), the presentation is clear, and experiments are well-designed and convincing, showing the final performance the community cares (i.e., the quality of the solution given fixed wall clock time). The authors also address many of the reviewers' concerns in the rebuttal, by including comparison with Gasse et al and different setting of the ILP solvers, further solidifying the work.

Overall, I happily accept the work. It can be a strong contribution to the learning-to-optimize community and I would highly encourage the authors to open source the code.